# Recent Progress in Surface Plasmon Resonance Biosensors (2016 to Mid-2018)

**DOI:** 10.3390/bios8040132

**Published:** 2018-12-16

**Authors:** Ewa Gorodkiewicz, Zenon Lukaszewski

**Affiliations:** 1Department of Electrochemistry, Institute of Chemistry, University of Bialystok, Ciolkowskiego 1K, 15-245 Bialystok, Poland; 2Faculty of Chemical Technology, Poznan University of Technology, 60-965 Poznan, Poland; Zenon.Lukaszewski@put.poznan.pl

**Keywords:** surface plasmon resonance, cancer markers, biosensors, receptor immobilization, antibodies, nanoparticles

## Abstract

More than 50 papers on surface plasmon resonance biosensors, published between 2016 and mid-2018, are reviewed. Papers concerning the determination of large particles such as vesicles, exosomes, cancer cells, living cells, stem cells, and microRNA are excluded, as these are covered by a very recent review. The reviewed papers are categorized into five groups, depending on the degree of maturity of the reported solution; ranging from simple marker detection to clinical application of a previously developed biosensor. Instrumental solutions and details of biosensor construction are analyzed, including the chips, receptors, and linkers used, as well as calibration strategies. Biosensors with a sandwich structure containing different nanoparticles are considered separately, as are SPR (Surface Plasmon Resonance) applications for investigating the interactions of biomolecules. An analysis is also made of the markers determined using the biosensors. In conclusion, there is shown to be a growing number of SPR applications in the solution of real clinical problems.

## 1. Introduction

Biosensors are the subject of enormous expectations and are gradually gaining in diagnostic importance. These expectations are connected with what is called “liquid biopsy” (i.e., diagnosis based on analysis of body fluids such as blood, urine, and saliva, and the possibility of early diagnosis of various cancers). However, there is still a shortage of biosensors offering near 100% sensitivity and specificity (i.e., 100% of true positive and 100% of correctly rejected results, respectively). An ideal biosensor should react exclusively to the target marker despite the presence of numerous similar proteins, glycoproteins, and others in the analyzed body fluid. Moreover, the biosensor’s dynamic response range should include the concentrations of the marker found in the body fluid, both of persons with the disease and of the healthy population. It is also expected that the precision of measurement of the marker concentration will be sufficient to distinguish between samples below and above a “cut-off” value.

A limited number of measuring techniques are used successfully in combination with biosensors, the leader among which is ELISA. Surface Plasmon Resonance (SPR) is still only a promising technique, which so far has no practical diagnostic applications. However, the number of potential applications of SPR in the solution of real clinical problems is growing.

This paper reviews the most recent publications on SPR biosensors, appearing between 2016 and mid-2018. Earlier works are covered in an excellent review by Masson (2017) [1]. This review excludes papers concerning SPR biosensors used for the determination of large particles such as vesicles, exosomes, cancer cells, living cells, and stem cells, as well as microRNA—these papers have already been broadly reviewed [2]. Both Masson and Ferhan et al. conclude that future work should focus more on clinical samples than on improving detection specificity and sensitivity.

## 2. Stages of Biosensor Development

Generally, a mature biosensor and a procedure for the determination of a particular marker are developed in several stages, beginning with the conception of the biosensor, followed by analytical characterization, validation, and determination of the marker in real samples. Therefore, the reviewed papers are categorized into five groups, depending on the degree of maturity of the reported solution. The following stages of development of a biosensor and a related analytical procedure can be distinguished:The biosensor is used only for the detection of a marker;The biosensor is characterized in terms of quantitative marker determination (calibration graph, the marker concentration range covered by the biosensor);The biosensor and related analytical procedure are validated (precision, recovery, interferences, comparison of results with another procedure such as ELISA, examples of natural samples, e.g., blood plasma);The mature biosensor and the analytical procedure are used for investigation of the marker in significant series of clinical samples, including long control series of healthy donors;A fifth stage may be distinguished when the mature SPR biosensor and procedure are used in clinical investigation.

This categorization proves to be useful in relation to the papers considered in this review. The papers pertaining to stages (i) and (ii) represent incomplete analytical procedures. Technical solutions applied in these biosensors are shown in Table 1 [3,4,5,6,7,8,9,10,11,12,13]. These papers represent high innovative potential in terms of biosensor construction, which may result in fully developed procedures in the future.

An example of a paper pertaining to stage (i) is a report on the detection of CEA in the blood of colon cancer patients [4]. CEA is captured from the blood serum by the mouse anti-CEA antibody immobilized on the biosensor surface. A significantly higher SPR signal is produced by colon cancer samples than by samples from healthy donors. However, that study did not include calibration or CEA quantification—in contrast to a paper by Chiu et al. [10], where quantification of CYFRA 21-1 in spiked human blood was performed on the basis of a calibration curve. CYFRA 21-1 is a circulating fragment of creatinine-19 which is a marker for cancer, including lung cancer. Due to the lack of validation of the developed biosensor, this paper pertains to stage (ii).

The papers pertaining to stages (iii) and (iv) represent complete analytical procedures [14,15,16,17,18,19,20,21,22,23,24,25,26,27]. These biosensors and related analytical procedures are ready to be subjected to clinical investigation for subsequent use in diagnosis. Additionally, biosensors in stage (iv) are characterized as potential disease markers by a series of measurements performed with clinically classified material. An example of a paper pertaining to stage (iii) is a report on the determination of PSA in human serum [14]. PSA concentration was quantified on the basis of a calibration curve. The developed biosensor and related analytical procedures were validated by parallel PSA determination using ELISA in series of blood serum samples spiked with PSA. Apart from analytical characteristics such as a calibration curve, precision determination and potential interferences, papers pertaining to stage (iv) contain the results of marker determination in series of clinically classified samples. An example is a paper on a biosensor and procedure for the determination of laminin-5 in human blood plasma [20]. This paper describes the optimization of parameters, as well as characteristics relating to precision, recovery and potential interferences. To validate the biosensor, laminin-5 was determined by ELISA and by the developed biosensor in two series of samples; the first containing samples from bladder cancer patients, and the second containing samples of healthy donors.

Technical solutions for biosensors representing completely developed procedures are shown in Table 2.

The number of biosensors placed in this category outnumber those with incomplete analytical procedures shown in Table 1. This is a positive observation and a good prognostic factor for future SPR biosensors.

Papers pertaining to stage (v) [28,29,30,31,32,33,34,35,36,37] represent completely developed analytical procedures that have been applied in various clinical investigations. An example is a paper reporting the determination of three markers, laminin-5, collagen IV, and matrix metalloproteinase-2, in the blood plasma of pediatric burn patients [32]. The methods for the determination of these markers had been developed and published previously, and the paper cited here merely reports an application of those solutions.

Technical solutions for developed biosensors applied in clinical investigations are shown in Table 3.

The numerous applications of previously developed biosensors provide evidence of their effectiveness. As Table 1, Table 2 and Table 3 show, a majority of the reviewed papers present mature solutions or are devoted to the application of previously developed biosensors in clinical investigations. This is a tendency corresponding to that recommended in the reviews by Mansson and Ferhan et al. [1,2], and gives an indication of the future growing significance of SPR.

## 3. Fluidic vs. Non-Fluidic Measurements

Approximately the same number of reviewed papers reported the use of fluidic and non-fluidic measurement. In two cases, the use of both options as alternatives was reported, and in five cases fluidic measurements were performed as microfluidic. There is a significant difference in the arrangement of a fluidic or non-fluidic measurement; a fluidic measurement is usually performed with in situ creation of a biosensor, while in the non-fluidic case the biosensor is prepared ex situ. In the fluidic version a biosensor is created during the measurement by sequential introduction of a linker, a receptor and a solution containing the determined marker. Measurement is performed with the biosensor in contact with solution. This is the major difference from the non-fluidic case. An example of fluidic SPR measurement is shown in Figure 1. Finally, the chip sensor is cleaned to prepare it for the next measurement. Subsequent measurements can be performed rapidly, as in flow-injection analysis.

A single biosensor usually contains several channels, processed simultaneously for multi-sample measurement or for processing the same solution to gain better precision. The volume of processed solution is the main difference between fluidic and microfluidic techniques; in the microfluidic case there is a tendency towards the miniaturization of the measuring process. Microfluidic measurement also uses an array of measuring points (see Figure 2 and [3]).

Non-fluidic measurement is usually performed in a stationary arrangement, with an array of separated measuring points. An example of non-fluidic measurement is shown in Figure 3. An array of measuring points is used for a single measurement to improve the precision of the result. Multi-sample measurement is also performed, as shown in Figure 3, as well as regeneration of the chip after measurement.

Non-fluidic SPR measurement is performed following gentle drying of the biosensor, which is the major difference as compared with fluidic measurements.

No information has yet been published concerning the comparison of fluidic and non-fluidic versions of SPR measurements. As regards instrumental solutions, both classical SPR and SPR imaging are equally represented; the imaging version is frequently associated with non-fluidic measurement.

## 4. Receptors

A crucial part of a biosensor is the receptor. The receptor must ensure that only the target marker is captured from the analyzed sample, as well as ensuring suitable effectiveness in terms of the strength of analytical signal sufficient for the determination of a marker in real samples. In the reviewed papers, appropriate antibodies were most frequently used as receptors. The antibody was attached to the gold chip surface via a linker. Most frequently, cysteamine (2-aminoethanethiol) was used as the linker. Cysteamine is fixed onto the gold surface by the thiol group, while the amine group is used for attachment of the antibody. An example of such receptor immobilization is shown in Figure 4.

An EDC/NHS protocol is applied for this purpose, with amide bond formation between the antibody’s carboxyl group and the linker’s amine group. Alternatively, MUA (11-mercaptoundecanoic acid) may be used in conjunction with the EDC/NHS protocol [4]. The amine group of the antibody is used for the junction. Similarly, HS-OEG-COOH [38], S2PEG6COOH [5] (where OEG and PEG denote oxyethylene subunits), and mercapto-propane sulfonate [8] have been used. In numerous papers, the commercially available CM5 chip with carboxylated dextran as the linker was employed. Another commercially available chip (Easy2Spot) is supplied in a form ready for antibody bonding [6].

Several solutions other than antibodies have been used as receptors. A marker’s inhibitor can be used as the receptor, as in the cases of the inhibitor ONX 0914 [23] and the inhibitor ARP 101 [24]; in both cases 1-octadecano-thiol was used as the linker, and the inhibitor was attached to the linker via hydrophobic interactions. An example of this type of receptor immobilization is shown in Figure 5. A receptor-imprinted polymer may also be used as the receptor [14,16].

A glass slide covered with gold is a typical chip base (see the example in Figure 4). Alternatively, a gold-covered glass prism is used. Usually, a chromium, or alternatively titanium, under-layer is employed. Only a few papers report other solutions, such as a gold-covered glass fiber [5], also with additional graphene layer [39], a gold nanohole array [3,40], or a prism covered with gold and graphene [41]. Frequently, a gold chip surface is covered by a polymer with holes, creating an array of free gold measuring points. A thick inert polycarbonate protective layer is also proposed [42].

## 5. Enhancement of the SPR Signal

As demonstrated by Brolo’s research group in Canada, periodic areas of nanoholes in a sandwich configuration may be used to enhance the SPR signal [3]. The first antibody captures the marker, while the marker captures the second antibody. The introduction of different nanoparticles in the sandwich can lead to much greater enhancement of the SPR signal. Table 4 summarizes the cases in which nanoparticles were applied. In the simplest solution, the first antibody attached by the EDC/NHS protocol captures the marker, which captures an aggregate consisting of a gold nanoparticle covered by a second antibody attached to the antibody surface by the same EDC/NHS protocol [38]. An example is shown in Figure 6. A similar solution is reported [43] in which the EDC/NHS protocol is used for the first antibody attachment, and a biotinylated antibody attached to streptavidin-decorated gold nanoparticles serves as the second. The preconcentration of the marker with magnetic microparticles covered by the antibody has been reported [44]. Finally, the signal is created indirectly by a selected aptamer released from the magnetic microparticles-antibody-marker-aptamer structure. A quantum dot having a CdSe and ZnS core or shell structure has also been used for SPR signal enhancement in a sandwich configuration [45], as have polydopamine-wrapped magnetic multi-walled carbon nanotubes [46].

## 6. Calibration Strategy

In our opinion, the most significant information concerning the calibration strategy is whether the range of measurable concentration covers the range of concentrations of the marker in cancer samples and a representative level for the healthy population. In some cases, the reported dynamic response range covers several orders of magnitude (e.g., [38]). However, linearity of the analytical response is obtained when the analytical signal is plotted against log marker concentration. In other cases, linearity of the analytical signal against the marker concentration is reported, but in a significantly narrower concentration range. Generally, all calibration graphs represent the Langmuirian curve type, in which the initial sections may approximately follow the strain lines, while the whole curve may be approximately linear when the analytical signal is plotted against log marker concentration (see Figure 7). Almost all of the reviewed papers containing calibration data report calibration on the basis of a linear calibration graph. This appears to be a reasonable choice, because determination of the logarithm of the marker concentration does not satisfy expectations for clinical results. An example of an application of straight-line calibration is the determination of collagen IV in the plasma of breast cancer patients [21]. The Langmuirian type calibration contains the initial linear section between 50 and 300 μg L^−1^. However, this relatively narrow concentration range was sufficient for the determination of collagen IV in the plasma samples of breast cancer patients as well as control samples from healthy volunteers. Only two papers refer to semi-log calibration [7,15]. The semi-log calibration curve of cytokeratin-19 from 0.01 to 100 pg L^−1^ is reported by Chiu et al. [7], with a detection limit equal to 1 fg L^−1^. However, the physiological level of the marker is 3 ng L^−1^, three orders of magnitude above the range of applicability of the biosensor. No attempt to determine cytokeratin-19 concentration in real samples was undertaken.

## 7. Markers

In the reviewed papers, the targets of the developed or merely applied biosensors are various types of cancer, as well as other diseases. These targets are listed in Table 5. Lung, bladder, and breast cancers are most frequently represented. Single papers have been devoted to colorectal, prostate, and head and neck squamous cell cancers, as well as acute leukemia. Among non-cancer applications, thermal injuries have been most frequently investigated, as well as acute myocardial infarction and acute appendicitis. Papers devoted to apoptosis, asthma, megaloblastic anemia, Parkinson’s disease, hypertension, primary renal disease, and diabetes have also been published. Some biosensors have found applications with several diseases; for example, 20S proteasome and UCHL1. In the majority of cases, the biosensors were used for the determination of markers in the blood serum or plasma, although in several cases urine was the target body fluid.

## 8. Molecular Interactions

Apart from the papers devoted to the determination of particular markers in body fluids, several papers on SPR describe investigations of molecular interactions. One of these [47] describes the interactions of immobilized Cancer Antigen 125 (CA 125) with several aptamers. Elsewhere, the interaction between recombinant Smurf2 protein and CNKSR2 protein was described [48]. Other studies have investigated the parameters of binding between galectin-3 and pectin [49] and the glycosylation-dependent binding of galectin-8 to activated leukocyte cell adhesion molecule (ALCAM) [50]. A further study [51] investigated the affinity and competitive inhibition of nine caffeoylquinic acid compounds (CQAs) against programmed cell death protein 1 (PD-1) and its ligand PD-L1. Another investigation concerned the binding affinities of prostate-specific antigen to six lectins [52], and elsewhere, the ability of nanobody-targeting VEGFR (NTV1) to bind VEGFR2 D3 was demonstrated [53]. Generally, the aim of these papers is to investigate anti-cancer drugs and therapies. For example, in the last case [53], the nanobody NTV1 (antibody part containing the antibody domain) is presumed to have antiangiogenic properties due to the blocking of VEGFR2, which is a significant element in the triggering of cancer angiogenesis. SPR measurements were used for the determination of the binding constant of NTV1 to VERFR2. This possibility is available only with classical SPR and the fluidic type of measurement. 

## 9. Conclusions

The reviewed papers are categorized into five groups, depending on the degree of maturity of the reported solution; ranging from simple marker detection to clinical application of a previously developed biosensor. A majority of the reviewed papers represent validated biosensors and related analytical procedures. Numerous papers report clinical investigations with cancer markers and other diseases as the targets of biosensors.

Some of the reviewed papers used fluidic measurement arrangements, while others used stationary non-fluidic measurement with an array of measuring points. In spite of the significant differences in the measuring process, both versions are successfully used in the determination of various markers. Unfortunately, no information has yet been published concerning the comparison of fluidic and non-fluidic versions of SPR measurements. As regards instrumental solutions, both classical SPR and SPR imaging are almost equally represented. The imaging version is frequently associated with non-fluidic measurement. The fluidic version in combination with classical SPR provides an opportunity to determine the binding constant of molecular interactions.

The selection of a biosensor receptor is mostly determined by the chemical structure of the marker to be analyzed. Antibodies are the most frequently used type of receptors, in combination with the EDC/NHS protocol applied for the covalent attachment of an antibody to the biosensor surface via a suitable linker. Occasionally, marker inhibitors or marker-imprinted polymers have been used as receptors.

Several approaches to marker signal enhancement have been reported. A sandwich structure—consisting of a first antibody, a marker, and an aggregate consisting of a gold nanoparticle covered by a second antibody—has been used, as has preconcentration of a marker with magnetic nanoparticles.

As regards calibration strategy, all calibration graphs represent the Langmuirian curve type, in which the initial sections may approximately follow the strain lines, while the whole curve may be approximately linear when the analytical signal is plotted against log marker concentration. Almost all of the reviewed papers containing calibration data report calibration on the basis of a linear calibration graph, although semi-logarithmic curves were also used.

Several papers have reported molecular interactions. The possibility of determining the binding constant of interacting molecules is available only with classical SPR and the fluidic type of measurement.

## Figures and Tables

**Figure 1 biosensors-08-00132-f001:**
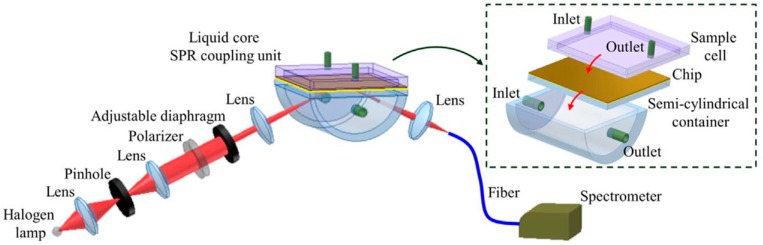
Example of fluidic SPR measurement. Reproduced with permission from [4]. Copyright (2018) Elsevier BV.

**Figure 2 biosensors-08-00132-f002:**
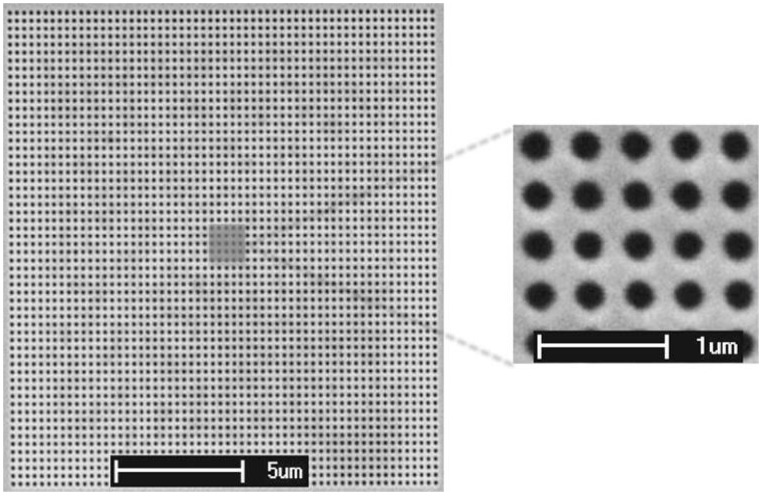
An array of gold nanohole measuring points used in microfluidic measurement. Reproduced with permission from [3]. Copyright 2015, Springer Science Business Media New York.

**Figure 3 biosensors-08-00132-f003:**
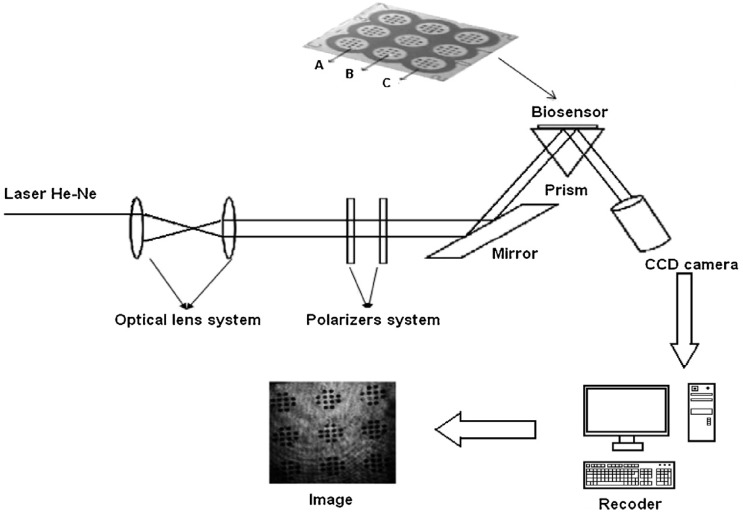
Example of non-fluidic measurement. Reproduced with permission from [27]. Copyright 2017 Elsevier B.V.

**Figure 4 biosensors-08-00132-f004:**
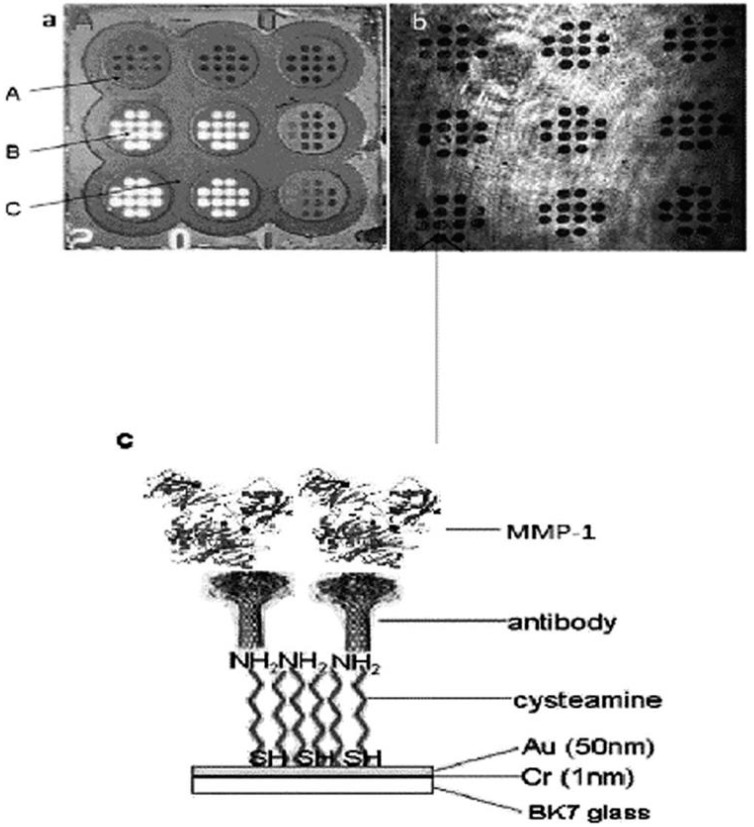
Example of covalent antibody immobilization (**c**) and picture of the chip (**a**) and image of the chip obtained by a CCD camera (**b**). Reproduced from [22] with permission of Royal Society of Chemistry.

**Figure 5 biosensors-08-00132-f005:**
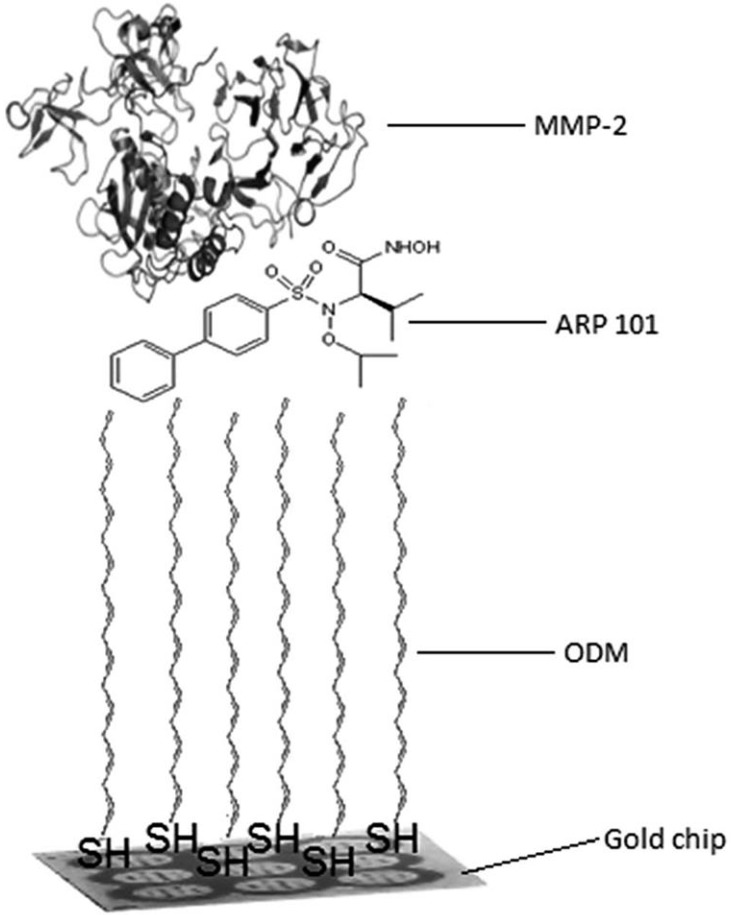
Example of receptor immobilization via hydrophobic interaction. Reproduced from [24] with permission of Royal Society of Chemistry.

**Figure 6 biosensors-08-00132-f006:**
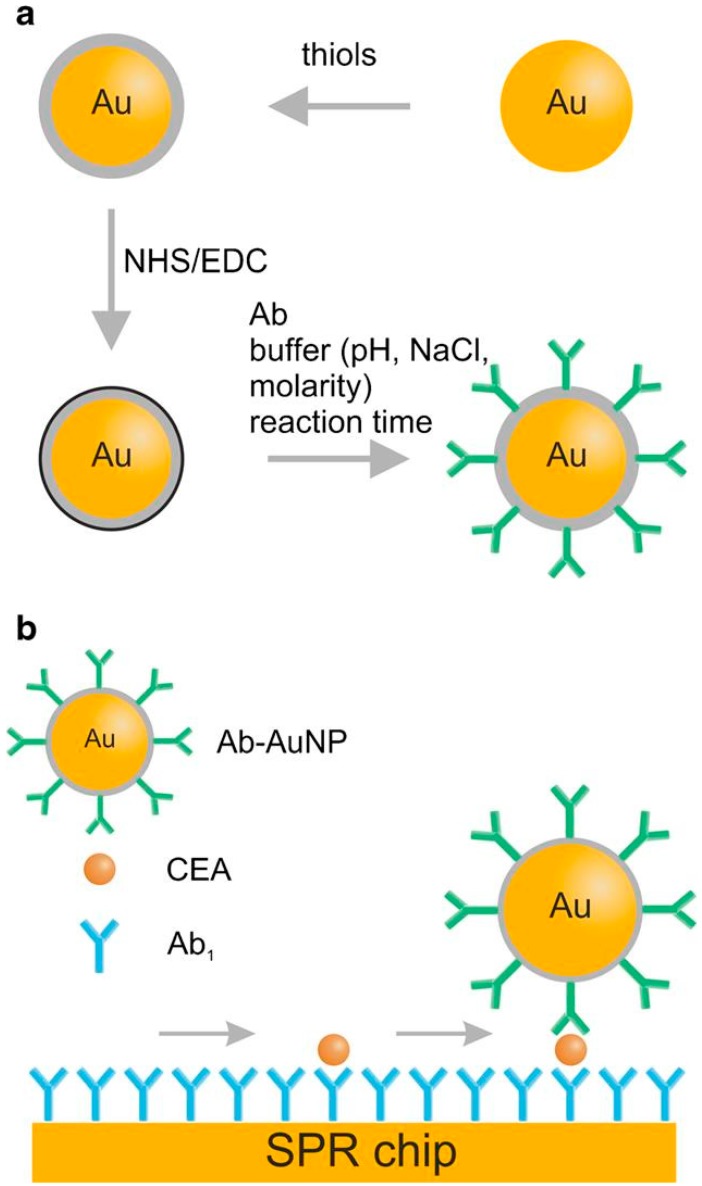
Example of the use of nanoparticles for SPR signal enhancement. (**a**) Scheme of nanparticles processing, (**b**) sandwich assay. Reproduced with permission from [38]. Copyright 2017, Springer-Verlag Berlin Heidelberg.

**Figure 7 biosensors-08-00132-f007:**
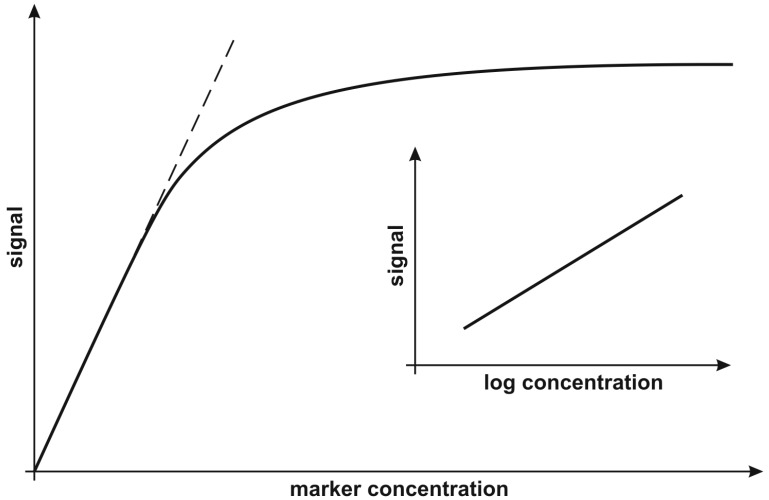
Typical calibration curves.

**Table 1 biosensors-08-00132-t001:** Technical solutions for biosensors in the initial stages of development.

Stage	Marker	SPR Type	Fluidic/Non Fluidic	Chip	Linker/Receptor	Receptor Immobilization	Reference
i	HER2	SPR	Micro-fluidic	Nano-whole array	Cysteamina/Sandwich/2Antibodies	Biotin/streptavidin	[3]
i	CEA	SPR	Fluidic	Slide/Cr/Au	MUA/antibody	EDC/NHS	[4]
i	Cytokeratin 17	SPR	Non fluidic	Optical fiber/Au	S2PEG6COOH/antibody	EDC/NHS	[5]
i	Cytochrom C	SPRi	Fluidic	Easy2Spot	Antibody	Sensor pre-activated G-type Senseye	[6]
i	BSA	SPR	Fluidic or Non fluidic	Prism/Au	Cysteamine/GOCOOOH/antibody	EDC/NHS	[7]
i	BSA	SPR	Non fluidic	Slide/Au	Mercapto-propane sulfonate/modified GO	EDC/NHS	[8]
i	BSA Biotin	SPR	Micro-fluidic	Nanogroove metasurface	None streptavidin	Physisorption/Streptavidin/biotin	[9]
ii	Cytokeratin-19	SPR	Fluidic or Non fluidic	Prism	Cysteamine/GOCOOOH/antibody	EDC/NHS	[10]
ii	Transferrin	SPR	Fluidic	Slide/Au	4-Mercapto phenylboronic	4-Mercapto phenylboronic	[11]
ii	Folic acid	SPR	Fluidic	Prism/Ti/Au/graphen	FAP	Hydrophobic interaction	[12]
ii	DNA *E. coli*	LSPR	Micro-fluidic	Slide/Ag nano prism	5′[C6-thiol] DNA probe	DNA probe-DNA target interaction	[13]

EDC/NHS—covalent amide bond formed due to the EDC/NHS protocol. EDS-1(ethyl-3-(dimethylaminopropyl)carboimide NHS-N-Hydroxysulfosuccinoimide, FAP-folic acid protein

**Table 2 biosensors-08-00132-t002:** Technical solutions for biosensors representing completely developed procedures.

Stage	Marker	SPR Type	Fluidic/Non Fluidic	Chip	Linker/Receptor	Receptor Immobilization	Reference
Iii	PSA	SPRi	Fluidic	Slide/Au	Allyl mercaptan	PSA imprinted polymer	[14]
Iii	CBP	SPR	Fluidic	CM5 chip	Dextran-COOH/Antibody	EDC/NHS	[15]
Iii	Troponin T	SPR	Fluidic	Slide/Au	Polydopamine/Epitop/	Polymer Imprinted epitop	[16]
Iv	Rac1, Rac1b	SPR	Fluidic	CM5 chip	Dextran-COOH/Antibody	EDC/NHS	[17]
Iv	5LOX	SPR	Fluidic	CM5 chip	Dextran-COOH/Antibody	EDC/NHS	[18]
Iv	CDK4	SPR	Fluidic	CM5 chip	Dextran-COOH/Antibody	EDC/NHS	[19]
Iv	Laminin-5	SPRi	Non fluidic	Slide/Au/array	Cysteamine/Antibody	EDC/NHS	[20]
Iv	Collagen IV	SPRi	Non fluidic	Slide/Au/array	Cysteamine/Antibody	EDC/NHS	[21]
Iv	MMP1	SPRi	Non fluidic	Slide/Au/array	Cysteamine/Antibody	EDC/NHS	[22]
Iv	20S immune-proteasome	SPRi	Non fluidic	Slide/Au/array	1-octadecano-thiol/Inhibitor ONX 0914	Hydrophobic interaction	[23]
Iv	MMP2	SPRi	Non fluidic	Slide/Au/array	1-octadecano-thiol/Inhibitor ARP 101	Hydrophobic interaction	[24]
Iv	YKL40	SPR	Fluidic	CM5 chip	Dextran-COOH/Antibody	EDC/NHS	[25]
Iv	Mortalin and αSynuclein	SPR	Fluidic	CM5 chip	Dextran-COOH/Antibody	EDC/NHS	[26]
Iv	Fibronectin	SPRi	Non fluidic	Slide/Au/array	Cysteamine/Antibody	EDC/NHS	[27]

EDC/NHS—covalent amide bond formed due to the EDC/NHS protocol.

**Table 3 biosensors-08-00132-t003:** Technical solutions for developed biosensors applied in clinical investigations.

Stage	Marker	SPR Type	Fluidic/Non Fluidic	Chip	Linker/Receptor	Receptor Immobilization	Reference
V	podoplanin	SPRi	Non fluidic	Slide/Au/array	Cysteamine/Antibody	EDC/NHS	[28]
V	20S proteasome	SPRi	Non fluidic	Slide/Au/array	Cysteamine/Inhibitor PSI	EDC/NHS	[29]
V	Cystatin C	SPRi	Non fluidic	Slide/Au/array	Cysteamine/Antibody	EDC/NHS	[30]
V	UCHL1	SPRi	Non fluidic	Slide/Au/array	Cysteamine/Antibody	EDC/NHS	[31]
V	MMP2 Laminin5 Collagen IV	SPRi	Non fluidic	Slide/Au/array	1-octadecano-thiol/Inhibitor ONX 0914 Cysteamine/antibody	Hydrophobic Interaction EDC/NHS	[32]
V	UCHL1	SPRi	Non fluidic	Slide/Au/array	Cysteamine/Antibody	EDC/NHS	[33]
V	20S proteasome	SPRi	Non fluidic	Slide/Au/array	Cysteamine/Inhibitor PSI	EDC/NHS	[34]
V	UCHL1	SPRi	Non fluidic	Slide/Au/array	Cysteamine/Antibody	EDC/NHS	[35,36]
V	20S proteasome	SPRi	Non fluidic	Slide/Au/array	Cysteamine/Inhibitor PSI	EDC/NHS	[37]

EDC/NHS—covalent amide bond formed due to the EDC/NHS protocol.

**Table 4 biosensors-08-00132-t004:** Stages of biosensor development and technical solutions for biosensors with the use of nanoparticles.

Stage	Marker	SPR Type	Fluidic/Non Fluidic	Chip/NP	Sandwich/Other	Receptors Immobilization (Chip/Antibody)	Reference
ii	Folic acid	SPRi	Non fluidic	Array/Cr/Au/FA-AuNP	Sandwich	HS-(CH2)11-EG3-NTA/polyhistidine	[12]
ii	Troponin I	SPR	Fluidic	Slide/Au/HGNP/MMWCNTs-PDA	Sandwich/MMWCNTs-PDA	Polydopamine/Polydopamine	[45]
iii	CEA	SPR	Fluidic	Slide/Ti/Au/AuNP	Sandwich	HS-OEG-COOH/HS-OEG-COOH/EDC/NHS	[38]
iii	HER 2	SPR	Micro-fluidic	Prism/Au/SAv-GNPs	Sandwich	MUA/EDC/NHS/biotinylated antibody	[43]
iv	Cytochrom C	SPR	Fluidic	Slide/Au/AuNR	Sandwich/MMP	Straptavidin/biotinylated aptamer/antibody/MMP	[44]
iv	AFP, CEA CYFRA 21-1	SPR	Micro-fluidic	Prism/Au	Sandwich/QD	Hexanedithiol/antibody/DTBE	[46]

SAv-GNPs—streptavidin decorated gold nanoparticles. FA-AuNP—AuNP functionalized with polyhistidine tagged folic acid binding protein. AuNR—Au nanorods. MMP—micro magnetic particles. QD—quantum dot (CdSe/ZnS core/shell structure). DTBE—2,2′-dithiobis [1-(2-bromo-2-methylpropionyloxy)]ethane. AFP—α-fetoprotein. MMWCNTs-PDA—polydopamine-wrapped magnetic multi-walled carbon nanotubes. HGNP—hollow gold nanoparticles.

**Table 5 biosensors-08-00132-t005:** Markers and related diseases.

Marker	Abbrev.	Cancer/or Other Disease	Body Fluid	Reference
Arachidonate 5-lipoxygenase	5LOX/ALOX5	Breast cancer	Blood plasma	[18]
Carcinoembryonic antigen	CEA	Colorectal cancer	Blood serum	[4]
Calcium Binding Protein	CBP	Acute myocardial infarction	Blood serum	[15]
Chitinase-3-like protein 1	CHI3L1/YKL-40	Asthma	Blood serum	[25]
Collagen IV		Breast cancer/burns	Blood serum	[21]
Cyclin-dependent kinase 4	CDK4	Lung, head and neck cancers	Blood serum	[19]
Cystatin C		Bladder cancer	Blood serum, urine	[30]
Cytochrom C		Apoptosis	No information	[6]
Cytokeratin 17	CK 17	Lung cancer	No information	[5]
Cytokeratin 19	CK19	Lung cancer	Blood plasma	[10]
Epidermal receptor protein-2 antigen	HER	Breast cancer	No information	[3]
Fibronectin		Burns	Blood plasma	[27]
Folic acid	FA	Megaloblastic anemia	blood	[12]
20S-immunoproteasom	20Si	Acute leukemia	Blood plasma	[23]
Laminin 5		Bladder cancer/burns	Blood plasma	[20,32]
Matrix metalloproteinase-1	MMP1	Bladder cancer/acute appendicitis	Blood serum	[22]
Matrix metalloproteinase-2	MMP2	Burns	Blood plasma	[24,32]
Mortalin/mitochondrial 70 kDa heat shock protein	mtHsp70	Parkinson’s Disease	Blood serum	[26]
Podoplanin		Bladder cancer	Blood serum, urine	[28]
20S-proteasom	20Sc	Burns, acute appendicitis, Cryptorchidism	Blood plasma	[34]
Prostate specific antigen	PSA	Prostate cancer	Blood serum	[14]
Ras-related C3 botulinum toxin substrate 1	Rac1	Non-Small Cell Lung Cancer	Blood serum	[17]
Ras-related C3 botulinum toxin substrate 1b	Rac1b	Non-Small Cell Lung Cancer	Blood serum	[17]
Transferrin	Trf	Hypertension, primary renal disease, diabetes.	Artificial urine	[11]
Troponin T	TnT	Acute myocardial infarction	Blood serum	[16]
Ubiquitin carboxyl-terminal hydrolase L1	UCHL1	Burns, cryptorchidism, Acute Appendicitis	Blood serum	[31,35,36]

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
