# Peer review of "Recent Progress in Surface Plasmon Resonance Biosensors (2016 to Mid-2018)"

_biosensors, 2018, doi:10.3390/bios8040132_

Reviewer 1 Report

The review reports literature reviews of surface plasmon resonance biosensors published in 48 literature. It seems that the paper is prepared in haste. It has left many flaws and inconsistency throughout the review.

The manuscript is too descriptive. It consists too many words and too little diagrams or results to support the claims.

The review that lacks experimental or theoretical results in it is considered weak (see, Ref [1]).

I am highlighting them below:

Title is misleading

Page 1, Line 35 Table Heading: Define what do the authors mean the term “stage” here.

The review misses a few representative diagrams related to SPR Biosensors.

Authors should break long tables into short ones. Create one table for each cancer stage to help assist the readers.

What do the authors mean by the term “both”? clarify it.

Authors are encouraged to clarify the difference between fluidic, non-fluidic, microfluidic and both.

Page 4, Line 41: Move table 1 above table 2. Authors should order the tables correctly.

Page 5, Line 53: Describe table first. Move on to the description afterwards.

Page 5, Line 54: No piece of information is considered trivial in science. In experimental studies, limits of detection and sensitivity of the biosensors are considered very important (see Ref [2, 3])

Page 6, Line 105: Provide citations for the gold chip covered by the polymer [2].

Page 6, Line 120: Replace the sentence by the following: As demonstrated by Brolo’s research group in Canada, periodic areas of nanoholes in a sandwich configuration may be used to enhance the SPR signal. Add new Figure 2 in the text. I encourage authors to cite Ref [4] and pot a Figure from this manuscript. Authors can either obtain permission for publishing or modify slightly and cite it. Depending on the MDPI policy, authors can state like this: Adopted from [4]. Also, see a recent paper by Meunier et al. as [5]

The manuscript uses the term “solution” at several locations. It seems that the solution here implies the methods used. The solution can also mean chemical composition in chemistry. Therefore, the solution term is misleading here. Replace the solution with an appropriate word where possible.

Include additional figure 3. The manuscript should consist of at least three figures. As stated above, manuscript without a result considered weak.

Page 7, Line 133: Can authors explain what they mean by solutions:

Chemical solutions, biological solutions, or methods.

Page 7, Line 138: Can authors give examples of other diseases here?

Page 7, Line 139: “Single papers have been devoted to apoptosis, asthma, megaloblastic anemia, Parkinson’s disease, hypertension, primary renal disease and diabetes.” This sentence doesn’t seem grammatically correct. “To the best of the authors’ knowledge, there are no single papers that have been devoted… “What do the authors’ mean by single papers? Clarify it.

[1] "Survey of the 2009 commercial optical biosensor literature," Journal of Molecular Recognition, vol. 24, no. 6, pp. 892-914, 2011.

[2] "Magneto-Optic Surface Plasmon Resonance Ti/Au/Co/Au/Pc Configuration and Sensitivity. Magnetochemistry 2018, 4, 35"

[3] "Improved magneto-optic surface plasmon resonance biosensors," in Photonics, 2018, vol. 5, no. 3, p. 15: Multidisciplinary Digital Publishing Institute.

[4] "Microfluidic plasmonic biosensor for breast cancer antigen detection," Plasmonics, vol. 11, no. 1, pp. 45-51, 2016.

[5]"Multiperiodic nanohole array for high precision sensing," Nanophotonics.

The manuscript is a good effort. However, it is not suitable for publishing as is. After carrying out a careful revision and all the above requested changes are carefully addressed, this manuscript should be considered suitable for publication. 

Author Response

We are obliged for the very helpful recommendations. Certainly, the original version was prepared hastily. The review has been completely restructured and its parts have been rewritten. The long Table 1 has been split into three tables. Six new figures have been introduced, including schemes of SPR biosensors and the recommended figure. Recommended citations have been added. We hope that the new version of the paper better corresponds to the title. The text has again been carefully checked by a native speaker. Generally, all of the Reviewer’s remarks have been addressed by changes to the manuscript. All changes are indicated in the revised manuscript.

Reviewer 2 Report

Type of manuscript: Review

Title: Recent progress in surface plasmon resonance biosensors

 (2016 to mid-2018)

Journal: Biosensors

This manuscript tries to report recent progress in surface plasmon resonance biosensors which were published in journal papers from 2016 to mid 2018. Authors try to categorize the published biosensors according to their features including types of linkers and receptors used, sample injection methodologies, protocols of bio-agent conjugation and etc. However, I believe that this manuscript should not be published in the journal due to the following serious reasons:

Authors never provide any scientific insight into development of SPR based biosensors but offer a list of short description of each biosensor published recently.

Authors do not follow a formally accepted format of a paper such as that consisting of introduction, main results and discussion, and conclusion but a series of paragraphs successively placed without reasonable grounds.

The tables that authors present in the manuscript do not lead to any scientifically meaningful guidelines and clues to which readers would likely refer for device progress and development.

Authors fail to highlight/focus on points worth being considered for SPR sensor device but try to give a list of techniques frequently adopted for conducting relevant experiments without addressing their important aspects.  

References do not seem to be appropriate in a scientifically general sense

Some of the sentences are hard to understand largely due to the improper use of English words, and misspelled ones.

Author Response

We are obliged for the review of our manuscript. We disagree with several of the points made. In our view, a review cannot have the structure of a regular paper, particularly an experimental part. However, the introduction and conclusions have been substantially expanded. As regards references, our paper reviews all papers concerning SPR published within a period from 2016 to mid-2018 (with the exception of recently reviewed papers concerning large particles and microRNA). Thus, the assertion of a poor selection of references appears unfounded. The text has been carefully checked by a native speaker. Despite the critical evaluation of our paper, the Reviewer’s comments have been helpful in the preparation of a revised version of the manuscript. We hope that the revised manuscript will meet the approval of the Reviewer.

Reviewer 3 Report

In this manuscript, the authors have reviewed 50 literatures reported on the research of surface plasmon resonance biosensors from 2016-2018. They have mainly focus the projects that related to the detection of small biomarkers. Different applications of SPR sensors have been introduced and summarized. The table in the manuscript can be used as a guidance for the audience to review the recent trends on this topic. I recommend the paper for publishing in this journal after the below concerns have been addressed:

Comments:
1. In Page 1, Line 12, the authors have stated that “The reviewed papers are categorized into five groups, depending on the degree of maturity of the reported solution: ranging from simple marker detection to clinical application of a previously developed biosensor.” However, it is very confusing to me that which of the five groups means? And what is the specific title for each section? There should be five-section title indicated in the context according to this statement.

2. There are three summarized tables in this manuscript listing several parameters from each literature, which is good for the audience to track the record. However, there are far too few words to explain and even to give an explanation on each table contents. There should be at least five papers to be extracted from each Table to give a detailed highlight on what and how the projects functioned. One or two paragraph should be given on each highlighted literatures.

3. Currently, the word number in this manuscript is way too short, just one to two page. There is no detailed description both in the introduction part and conclusion part. Both of the two sections should be expanded to form a complete manuscript. No specific challenges and their corresponding solutions have been discussed and proposed in the conclusion part.

4. The paper has no section separation. It consists only a mixture of all the information and the present form is very difficult for the audience to obtain useful knowledge through the one or two single paragraph. Break the information into several subsection is required. The authors should make a division of each section, and give each section a title.

5. Before and after the introduction of the summary table, the authors should give detailed information extracted from the tables. The audience barely can get information from the context besides the Tables. Again, the context in the manuscript should help the audience to get more information out of the Tables, not the repeated information from the Table. For example, in Page 8, Line 159, the authors stated that “Generally, a majority of the reviewed papers represent validated biosensors and related analytical procedures, which means a shift towards the requirements of medical diagnostics. Evaluation of the results in these procedures is generally based on straight line calibration curves. Approximately a half of the papers are devoted to cancer markers, while fluidic and non-fluidic instrumental solutions appear with roughly equal frequency.”

Then why “there is a shift towards the requirements of medical diagnostics”? And how to shift? What are the specific challenges to this shift? “while fluidic and non-fluidic instrumental solutions appear with roughly equal frequency.” What are the advantages and disadvantages of the fluidic and non-fluidic instrumental solutions individually?

6. Although the authors have reviewed 50 papers in this paper, they miss several current literature references in their citation. For example, “Rapid real-time recirculating PCR using localized surface plasmon resonance (LSPR) and piezo-electric pumping" Lab on a Chip 17, 2821-2830, 2017. “Hybrid graphene/gold plasmonic fiber-optic biosensor" Advanced Materials Technologies-Wiley 2, 1600185, 2016. "Multifunctional hyperbolic nanogroove metasurface for submolecular detection" Small 13, 1700600, 2017.

Author Response

We are obliged for the very helpful recommendations. Following the recommendation, the paper has been restructured with numerous subtitles, unclear sections have been rewritten, and six new figures have been introduced. The Introduction and Conclusions have been expanded. The recommended citations have been used. The text has again been carefully checked by a native speaker. Generally, all of the Reviewer’s remarks have been addressed by changes to the manuscript. All changes are indicated in the revised manuscript.

Round  2

Reviewer 1 Report

The manuscript is sufficiently revised. However, there are still some typos in the text as well as in the references. For example, see, Ref [42]. The quality of the manuscript is improved.

Author Response

Thank you again for the very valuable recommendations. We are confident that the quality of the revised paper is significantly improved. We have corrected reference [42], as well as similar inconsistencies in references [1], [3], [25] and [36].

Reviewer 2 Report

I believe that the manuscript has been approved a great deal except the conclusion section.

The conclusions made in the section are still in an unacceptable quality. The section needs to give more insightful direction with graspable points summarized. For this kind of a review paper, authors should be able to write a conclusion section with great care.

I would recommend the publication of this manuscript after major correction of the conclusion section.

Author Response

We are obliged for the valuable review of our revised paper.

We have to agree with the Reviewer’s opinion that the ‘Conclusions’ should be rewritten. We have prepared a much more extensive text which, we hope, will satisfy the Reviewer.

Reviewer 3 Report

The manuscript has been improved after revisions according to the reviewers' comments. However, there are a few points still need to be addressed before it is published.

 1. The authors have not replied my Comment 2 very well.

“There should be at least five papers to be extracted from each Table to give a detailed highlight on what and how the projects functioned. One or two paragraph should be given on each highlighted literatures. “ The selected literatures need to be described in detail with scientific parts systematically introduced in the context.

 For example,

In Page 15, in the section of “Molecular interactions”,

“One of these [47] describes the interactions of immobilized Cancer Antigen 125 (CA 125) with several aptamers. Elsewhere, the interaction between recombinant Smurf2 protein and CNKSR2 protein was described [48]. Other studies have investigated the parameters of binding between galectin-3 and pectin [49] and the glycosylation- dependent binding of galectin-8 to activated leukocyte cell adhesion molecule (ALCAM) [50].”

This is not scientific description for the reference, it seems that the authors have not even gone through the whole paper of each reference, just the title. One paper for 200-300 words is the minimum requirement, for introducing the setups, the surface functionalization schemes, for the novelty of the reported approach compared to the conventional one, etc. 3-5 papers for each topic should be described in this scientific way.

 2. In the conclusion part,

 “Conclusions

A majority of the reviewed papers represent validated biosensors and related analytical procedures. Numerous papers were devoted to clinical investigations with cancer markers and other diseases as the targets of biosensors. The calibration strategy was generally based on straight-line calibration curves, although semi-logarithmic curves were also used. Antibodies were the most frequently used type of receptors. Some of the reviewed papers used fluidic measurement arrangements, while others used stationary non-fluidic measurement with an array of measuring points.”

 What the authors gave is just a summary of the trend of the paper. Thers is no advantages and disadvantages of each SPR technique be given. For researchers, they do not need this kind of information. They need the guideline for the design and selection of the SPR technique, according to their specific requirements or goals. The review paper should give the guideline for the researchers to solve the technical problem.

Author Response

Reviewer 3

We are obliged for the valuable review of our revised paper.

We have to agree with the Reviewer’s opinion that the ‘Conclusions’ should be rewritten. We have prepared a much more extensive text which, we hope, will satisfy the Reviewer.

However we partly disagree with the suggestion contained in the comment:

There should be at least five papers to be extracted from each Table to give a detailed highlight on what and how the projects functioned. One or two paragraph should be given on each highlighted literatures. The selected literatures need to be described in detail with scientific parts systematically introduced in the context.

 The recommended approach would mean that 25 papers out of a total of 50 should be reviewed in detail. This is a different concept of the article than our own.

We prefer to extract from all of the reviewed papers significant problems such as the maturity of the developed biosensor from the point of view of clinical application, operational differences, selection of calibration strategy, etc. However, we agree that illustration of the text with examples is a correct recommendation. Therefore, five examples of approximately 100 words have been added (two on page 3, one on page 4, one on page 15 and one on page 17), in addition to the completely rewritten ‘Conclusions’.

Round  3

Reviewer 2 Report

The conclusion section has been improved enough for this manuscript to be acceptable for publication in the journal.

Author Response

We are very obliged for the review of our paper. The text has again been carefully checked by a native speaker.

Reviewer 3 Report

The manuscript has been improved after authors' revision.

Author Response

(The authors gave the same response as above.)
